# Visual Perception and Fixation Patterns in an Individual with Ventral Simultanagnosia, Integrative Agnosia and Bilateral Visual Field Loss

**DOI:** 10.3390/neurolint17070105

**Published:** 2025-07-10

**Authors:** Isla Williams, Andrea Phillipou, Elsdon Storey, Peter Brotchie, Larry Abel

**Affiliations:** 1Department of Medicine Alfred Hospital, Monash University, Melbourne 3004, Australia; 2St Vincent’s Hospital, Melbourne 3065, Australia; 3Orygen, Melbourne 3052, Australia; andrea.phillipou@orygen.org.au; 4Centre for Youth Mental Health, The University of Melbourne, Melbourne 3010, Australia; 5Department of Psychological Sciences, Swinburne University of Technology, Melbourne 3122, Australia; 6Department of Mental Health, St Vincent’s Hospital, Melbourne 3065, Australia; 7Department of Mental Health, Austin Health, Melbourne 3084, Australia; 8Department of Radiology, St Vincent’s Hospital, Melbourne 3065, Australia; brp@unimelb.edu.au; 9Optometry, School of Medicine, Faculty of Health, Deakin University, Geelong 3216, Australia; 10Department of Optometry and Vision Sciences, University of Melbourne, Melbourne 3010, Australia

**Keywords:** simultanagnosia, scanpaths, perception, face processing simultanagnosia

## Abstract

Background/Objectives: As high-acuity vision is limited to a very small visual angle, examination of a scene requires multiple fixations. Simultanagnosia, a disorder wherein elements of a scene can be perceived correctly but cannot be integrated into a coherent whole, has been parsed into dorsal and ventral forms. In ventral simultanagnosia, limited visual integration is possible. This case study was the first to record gaze during the presentation of a series of visual stimuli, which required the processing of local and global elements. We hypothesised that gaze patterns would differ with successful processing and that feature integration could be disrupted by distractors. Methods: The patient received a neuropsychological assessment and underwent CT and MRI. Eye movements were recorded during the following tasks: (1) famous face identification, (2) facial emotion recognition, (3) identification of Ishihara colour plates, and (4) identification of both local and global letters in Navon composite letters, presented either alone or surrounded by filled black circles, which we hypothesised would impair global processing by disrupting fixation. Results: The patients identified no famous faces but scanned them qualitatively normally. The only emotion to be consistently recognised was happiness, whose scanpath differed from the other emotions. She identified none of the Ishihara plates, although her colour vision was normal on the FM-15, even mapping an unseen digit with fixations and tracing it with her finger. For plain Navon figures, she correctly identified 20/20 local and global letters; for the “dotted” figures, she was correct 19/20 times for local letters and 0/20 for global letters (chi-squared NS for local, *p* < 0.0001, global), with similar fixation of salient elements for both. Conclusions: Contrary to our hypothesis, gaze behaviour was largely independent of the ability to process global stimuli, showing for the first time that normal acquisition of visual information did not ensure its integration into a percept. The core defect lay in processing, not acquisition. In the novel Navon task, adding distractors abolished feature integration without affecting the fixation of the salient elements, confirming for the first time that distractors could disrupt the processing, not the acquisition, of visual information in this disorder.

## 1. Introduction

When viewing a scene, we can readily decompose it into ever-finer elements until reaching the limit of our visual acuity. When asked, we can report back either the entire scene or some subtle detail within it. However, this ability can fail. In 1924, Wolpert introduced the term “simultanagnosie” to describe the inability of a patient to see “the whole simultaneously while having a good grasp of the details”, which he called “a disturbance of overall view” [1].

Later studies [2,3,4] found that there were clear differences in the nature and severity of the perceptual abnormalities present. Farah and colleagues separated them into dorsal and ventral simultanagnosia after the sites of the lesions, which are characteristic of each [5,6,7]. They noted that both groups could recognise simple shapes but struggled with interpreting complex scenes. They further observed that the fundamental perceptual deficits and lesion locations differ under these conditions—that the bilateral occipito-parietal lesions seen in dorsal simultanagnosia lead to a deficit in multiple object recognition and a failure to appreciate the spatial relationship between objects in a scene, while lesions of the left temporal-occipital cortex lead to a general slowing of information processing and prosopagnosia. Farah noted that with ventral lesions, patients can see multiple objects simultaneously but process them sequentially, with reading inability being especially prominent [6]. Patients with ventral simultanagnosia who have left temporo-occipital lesions can see and recognise multiple objects, albeit slowly, unlike patients with dorsal simultanagnosia. The dorsal form has been described as leading to deficits in the processing of multiple objects or multiple features of a single object, while the ventral form leads to reading impairment and a deficit in the processing of complex scenes [8]. Thus, while the two classes of simultanagnosia share features, they differ in their specifics. Some tasks may be impossible in dorsal simultanagnosia, while in the ventral form, simple, but not complex, versions of the same task may be possible.

Riddoch and Humphreys [9] described a patient with bilateral damage to the occipito-temporal cortex who could not integrate local parts of a figure into a higher-order shape despite having normal elementary sensory functions such as visual acuity, colour vision, brightness discrimination, and normal stored knowledge of objects. They called this integrative agnosia and noted that it might be a result of simultanagnosia. Similarly, when shown a large letter composed of smaller letters (a Navon letter), patients with dorsal simultanagnosia fail to identify the larger, global letter while readily seeing the small, local ones [10,11].

Because simultanagnosia is defined by deficits in processing multiple or complex images, the question obviously arises as to whether this is related to deficits in acquiring visual information in the first place; that is, do patients’ eyes fixate normally on elements of the visual scene they wish to identify? In a study of two patients with dorsal simultanagnosia, it was found that when they were asked to identify the global form of Navon letters (large letters composed of multiple small letters) [12], gaze patterns did not differ between successful and unsuccessful trials [13]. They identified 100% of the local letters but only 12/42 of the global ones, even though their scanning often covered the extent of the global letter [14]. A patient with ventral simultanagnosia arising from posterior cortical atrophy was also found to be severely impaired in recognising the global letter in the Navon test [15]. Their gaze behaviour was not recorded.

Another stimulus where identification of the global form depends on the integration of local features is pseudoisochromatic (e.g., Ishihara) colour plates, which test for colour vision defects by requiring patients to read out digits defined by similarly coloured circles surrounded by circles differing in colour. A report of seven patients with dorsal simultanagnosia found that they were unable to identify the digits but could correctly identify the colours of the individual circles [16]. In contrast, a patient described as having “profound simultanagnosia” arising from bilateral dorsal stream pathology had normal results on the Ishihara test [17]. A patient with ventral simultanagnosia was able to correctly identify single colours but was unable to integrate the coloured circles of the Ishihara plates [4]. In addition, he was unable to trace the digits on the plates with his finger.

A more familiar object with important features at both local and global scales is the human face. Faces convey both identity and emotion. One can even have an autonomic response to a familiar face that is consciously unrecognised, as has been described in prosopagnosia [18]. We might thus expect differences in how identity and affect are processed in simultanagnosia. There is also a well-established “inverted triangle” scanning pattern generated by normal individuals [2], which may be distorted in many conditions, e.g., posterior cortical atrophy (PCA) [2,19]. In other disorders (e.g., schizophrenia), the scanning pattern may be abnormal during undirected free viewing but may normalise during a task such as emotion recognition [20]. Studies on how faces are perceived in simultanagnosia are scarce. A patient with ventral simultanagnosia cannot recognise faces per se but can use features such as hair colour to identify individuals [4]. In contrast, a patient with dorsal simultanagnosia can recognise famous faces and extract gender and emotional information from them [21]. As areas specialised for face processing lie in the ventral portion of the temporal lobe [22], these differences in performance may be explained by the differences in neuropathology between the two categories of simultanagnosia. Another patient, described as having ventral simultanagnosia and dorsal prosopagnosia, could recognise famous faces but had difficulty discriminating between unfamiliar faces and identifying emotional expressions [23].

Although how fixations are distributed across a scene may play a significant role in how individuals extract information from it, very few studies of simultanagnosia, especially the ventral form, have recorded eye movements. One case report of a patient with simultanagnosia and damage to the occipital, parietal and temporal lobes, thus encompassing both the dorsal and ventral pathways, fixated the informative regions of the “Cookie Theft” test image but could only describe individual features of the scene, not the narrative the image represented [24]. For either form of simultanagnosia, studies reporting gaze behaviour have only reported such performance on a single task. Assessing gaze behaviour and perception across a range of tasks would allow us to better delineate the limits on our patient’s ability to integrate visual information and to determine what role, if any, abnormal information acquisition plays in these limits.

Here, we report on a patient with ventral simultanagnosia and integrative agnosia with ventral encephalomalacia of the occipital lobes due to bilateral occipital lobe infarction. She underwent clinical assessment and extensive neuropsychological evaluation, along with structural magnetic resonance imaging (MRI) evaluation. Her observation that she was unable to recognise faces and her failure to identify the Ishihara plates during clinical assessment led to those tasks’ inclusion in the testing protocol presented here. The emotional faces were added based on her description of how she made this judgement. The Navon figures were included, as they are by now a classic way to assess the integration of local information into a global unit. This formed a battery of perceptual tasks whose completion required holistic processing of local features: facial identification, facial emotion identification, and Navon figures, both in their usual form and when surrounded by black filled circles (“dotted” Navons), and pseudoisochromatic plates. They varied in the degree to which they depended on local or global processing. For each, gaze was recorded and analysed to determine if fixation patterns differed for successful and unsuccessful performance, particularly fixation of the salient elements of the stimuli. Facial recognition requires holistic processing of individual features. Emotion recognition is facilitated by the integration of several local features for some emotions, but for others, it depends largely on one feature. Both the global Navon figures and the characters in the pseudoisochromatic plates depend on the integration of simple local elements to be perceived. In addition, both the colour plates and the “dotted” Navon figures contained distractor local features, which could hinder the integration process. The use of this test battery allowed us to look for commonalities in performance across the tasks, and the recording of gaze allowed us to determine whether failures on the tasks were accompanied by changes in how visual information was acquired. We hypothesised that poor perceptual performance would be reflected in fixation of non-salient regions of the stimuli. Such studies are rare for dorsal simultanagnosia and virtually absent for the ventral form.

## 2. Materials and Methods

MH underwent several types of assessments. She underwent a full neuro-ophthalmological examination, a neuropsychological assessment, structural magnetic resonance imaging (MRI) and an evaluation of her eye movements as she performed a series of perceptual tasks. These were carried out initially with a Tobii 1750 eye tracker (Tobii Technology AB, Stockholm, Sweden), which operated at 50 Hz with an accuracy of 0.5 deg and a resolution of 0.25 deg. Recordings were made with a viewing distance of 75 cm. Others were carried out using an Eyelink 1000 Plus (SR Research, Ottawa, ON, Canada) after it became available. It operated at 500 Hz, with an accuracy of 0.25 deg and a resolution of 0.01 deg. The methodology for each task, including the tracker used, will be presented separately. As the eye movement analyses were of fixations rather than peak velocity or latency, we considered that, despite the grossly different frame rates, the two systems were comparable for the purpose of this study.

### 2.1. Famous Faces

Methods: Whilst a few cases of face recognition deficits in ventral simultanagnosia have been reported [3,4], none have included eye tracking, so it is unknown whether impaired recognition is associated with abnormal acquisition of visual information, as has been observed in prosopagnosia [25]. Using the Tobii 1750, we presented 10 black and white images of famous individuals from entertainment, politics and sport: Donald Bradman, Cary Grant, Marilyn Monroe, Marlon Brando, Humphrey Bogart, Audrey Hepburn, Elizabeth Taylor, Robert Menzies, Queen Elizabeth II, and President John F Kennedy, looking as if they would have before our patient suffered her initial incident. They were presented for 5 s each. There was no fixed time limit on the participant’s verbal response. For analysis, we drew areas of interest (AOIs) around the eyes, nose, and mouth. Tobii Clearview 2.7.1 software was used to identify the number of fixations of each AOI and the duration of each fixation.

### 2.2. Emotional Faces

Methods: In addition to identity, faces convey emotion. It has been proposed, although not universally accepted, that the mechanisms underlying the perception of identity and emotion are to some degree separable (see [26] (Calder & Young, (2005) for a review). We therefore tested our patient on the recognition of a standardised set of faces (https://www.paulekman.com/product/pictures-of-facial-affect-pofa, accessed on 10 June 2024) that expressed the emotions anger, disgust, fear, sadness, happiness, surprise and neutrality. Each emotion was represented by 8 different images (4 male, 4 female), with images being presented for 8 s, each preceded by a fixation cross. Images were presented on a 17” screen with a resolution of 1024 × 768, 90 cm away from the participant, who rested on a chinrest. The stimuli subtended 8 × 13 deg at the viewing distance from the eye. Following the presentation of each face stimulus, a list of all the emotions was presented on the screen, and MH was asked to verbally identify the emotion portrayed in the previous image. She was given as much time as needed to respond. Eye movements were recorded by an Eyelink 1000 (SR Research, Ottawa, ON, Canada). As before, areas of interest were defined for the eyes, nose and mouth, and the number of fixations and fixation duration were calculated for each area.

### 2.3. Colour Plates

Methods: Digital versions of ten Ishihara colour test plates were used as stimuli. Eye movements were recorded on a Tobii 1750, with a viewing distance of 75 cm. The colour plates subtended 15 deg of visual angle. The subject was given 15 s to identify the number displayed. On two plates, she was subsequently also given the opportunity to trace the digit with the tip of a pencil, having 1 min to do so.

### 2.4. Navon Figures, Plain and “Dotted”

Methods: To explicitly examine global versus local feature identification in our patient, stimuli consisting of larger letters composed of smaller letters were developed [12]. In all instances, the large and small letters were incongruent. Twenty characters per test set were presented, subtending, with slight variations between letters, 6 × 5 deg. The first set consisted of conventional Navon-type letters; the second used similar letters, but these were now surrounded by black circles of the same size. This mimicked the organisation of Ishihara plates, where the characters to be identified were surrounded by a field of coloured circles. We hypothesised that the increased visual complexity of the stimulus would disrupt the participant’s ability to integrate the local target letters into a global construct and that this would be reflected by changes in their fixation distribution. Each image was presented for 5 s, with the subject being asked to identify both the large and the small letters. The plain and “dotted” targets were intermixed pseudo-randomly. Stimuli were presented on a 17” screen with a resolution of 1024 × 768, 90 cm away from the participant, who rested on a chinrest, and eye movements were recorded by the Eyelink 1000 Plus, with fixations categorised as salient on the small letters and non-salient if they were in the surrounding region.

## 3. Results

### 3.1. Case Report

In 1975, MH, aged 33, fell backwards on the kitchen floor, bumping her head. She developed severe, continual occipital headache and pain on the right side of her neck. Six months later, MH felt vague and giddy and could not negotiate a gate. A short time later, she experienced a loss of consciousness and fell. When she woke in the hospital, her recollection was that “all was black”. A CT scan of the brain in 1975 revealed no abnormalities. MH was advised that she had hysterical blindness, but two months later, the examination revealed bilateral homonymous congruous visual field defects characteristic of bilateral occipital lobe infarction (Figure 1). Figure 2 shows that they have changed little over the years.

The visual acuity of each eye for distance was 6/12. Colour vision was intact when tested with the Farnsworth–Munsell 15D test, but MH could not read the Ishihara colour chart. This suggested a diagnosis of simultanagnosia. The optic discs were slightly pale. No further abnormal signs were found in the examination of the nervous system. The cerebral angiogram was normal. The management comprised antiplatelet therapy, antihypertensive medication and, in recent years, treatment of diabetes mellitus. An MRI scan confirmed bilateral occipital lobe infarction when it became available eleven years after the onset of her symptoms (Figure 3).

### 3.2. Neuropsychological Report

The examination of MH showed no evidence of visual neglect on either the multiple line division task [27] or the RA-cancellation task [28]. In the latter, she utilised a systematic “up and down” search strategy, beginning on the left. She performed confidently and quickly on Visual Form Discrimination and Judgement of Line Orientation tests [29].

Her object recognition/naming was assessed via the Boston Naming Test (2nd Edn) [30]. She correctly named ^38^/_60_ objects spontaneously and a further ^13^/_17_ with semantic (“stimulus”) clues for a total of ^48^/_60_ (1^3^/_4_ SD down). Qualitatively, several responses were of interest. She initially called item 2 (a tree) “a flower arrangement in a vase” until she saw the trunk and corrected herself. She initially complained that the drawing of a helicopter (item 11) “has too many lines”. She called a globe (item 27) “a fan” before correcting herself after a semantic clue, commenting that “the lack of colour doesn’t help”. She correctly identified 6 of the last 10 (lowest frequency) objects spontaneously, with a further 3 on phonemic cueing. There thus did not appear to be severe anomia, and indeed, there was no evidence of word-finding difficulty or dysfluency in the general conversation. In contrast, she had great difficulty with figure/ground discrimination on an overlapping figures test [31], describing it as “just a lot of lines”. When informed that one particular figure was a collection of fruit, she was able to pick out a banana, a pear, an apple, and a bunch of grapes (missing the halved orange), although this was a very slow process, taking several minutes. She also performed poorly on the CAMDEX [32] pictures of common objects photographed from unusual angles (a test of perceptual categorisation), misidentifying ^3^/_6_ (purse, cup of tea, and telephone).

MH described difficulty recognising people known to her, explaining that she could see people’s outlines but could only perceive one feature at a time. For instance, she could see the examiner’s moustache or his glasses, but not both together. She explained that this was not ameliorated by distance, as only single features were perceived, whether from 1 foot or 20. This indicates that the perceptual deficit does not arise from her restricted visual fields. She had learned to recognise people by hair colour, voice, stance/gait, and clothing. This suggested a form of prosopagnosia. This was investigated further with the Benton Test of Facial Recognition, which is in fact a test of facial matching. In the first section of the test, a face must be matched to an identical face in an array of five other, non-matching faces. MH performed perfectly on this section. In the second section, three of the six faces in the array are of the same person as the target face but are photographed from different angles or in different lighting. MH found this “much harder to do” and performed at a chance level.

MH was able to name displayed colours directly, state appropriate colours for various objects, and pick the appropriate coloured pencils to colour given named objects. In other words, she did not appear to have a colour agnosia (of any form). Her colour perception/matching was normal, as judged by Farnsworth’s Dichotomous Test for Color Blindness (D-15) (The Farnsworth Dichotomous Test for Color Blindness, Panel D-15: Manual—https://colormax.org/farnsworth-d-15-color-vision-test, accessed 6 July 2025) (available from ProTech Ophthalmics, Brentwood, CA, USA). However, she was unable to read the Ishihara pseudoisochromatic plates (even the orange-on-blue test plate), although she could describe the colours. She complained that the plates looked “dotty”. This difficulty did not improve with the plates at 3 metres, again showing that the perceptual deficit did not arise from her field deficit.

Simultanagnosia was investigated further with the Cookie Theft picture from the Boston Diagnostic Aphasia Examination (available from https://www.proedinc.com/Products/11850/bdae3-boston-diagnostic-aphasia-examinationthir.aspx, accessed 6 July 2025). Her description was as follows: … “there’s a cookie jar there … um … in a cupboard …with um … I thought it was a mother but it’s not—I think it’s a child … standing on a stool … and there’s another child … with its hand up … “(etc.). (Incidentally, it will be noted that she began describing the picture at the left-hand side—another illustration she does not suffer from left hemi-neglect.) She then transferred her gaze to the right side of the page: …”oh, it must be a kitchen … on the other side there’s a mother … and she is doing the dishes I think … and there’s a window with a curtain over it …”. She was unable to work out what the water flowing from the sink was, although she spent some time trying to decipher it. When asked for a summary of the picture, MH said: “it’s in the kitchen, with the children trying to get the cookies behind mum’s back”. In subsequent eye tracking studies, she was presented with this image for 2 min. Her scanpath (Figure 4) is consistent with her description of the scene. Note the numerous fixations of the overflowing water and of the task-irrelevant billowing curtain on the right.

MH’s largely intact single object recognition, as demonstrated by her Boston Naming Test (available from https://www.proedinc.com/Products/11870/boston-naming-testsecond-edition.aspx, accessed 6 July 2025) performance, was accompanied by largely intact copying of geometric shapes, on which her performance was ^3^/_4_ SD down [29]. Her copy of the Rey Complex Figure [33] was just over 1 SD down for accuracy for her age but was notably poorly organised. She had great difficulty performing the task, placing one finger on a feature of the original (black and white) diagram and another on her copy, constructing it in a piecemeal fashion, as shown. She commented that “I can’t see everything at once”. She was able to draw a distorted but recognisable bicycle to command and a clock face with the numbers and hands correctly positioned, although when drawing the circle, she explained that “I couldn’t see where I started”.

In contrast to MH’s complaint that she could only make out three or four letters at a time when reading and was consequently slow, her writing to dictation was normal. Her reading demonstrated the typical speed advantage of shorter words over longer words seen in this disorder.

At a later visit in 2004, MH’s anterograde episodic verbal memory was assessed via the Rey Auditory Verbal Learning Test (available from https://paa.com.au/product/ravlt/, accessed 6 July 2025). Her performance was generally superior, with her recall across the five learning trials being over 2 SD above average for her age and sex. There was no evidence of fragility of encoding (interference from the distractor list) or of abnormal forgetting across the delay period. Indeed, her delayed retrieval efficiency was 1 SD above average, and her recognition memory was almost perfect. Her verbal immediate memory was average, as illustrated by her forward digit span of 7, while her working memory (assessed by her backward digit span) was mildly impaired at 4.

### 3.3. Structural MRI Report

An MRI scan of the brain revealed multiple areas of encephalomalacia involving both the occipital and posterior medial temporal lobes, which were larger on the right, including the fusiform and lingual gyri bilaterally, the left superior cerebellum, and small lacunar infarcts within the thalami bilaterally. Within the temporal lobe, the damage involved the right lateral and medial occipito-temporal gyrus, the posterior body of the right hippocampus, and both parahippocampal gyri posteriorly (Figure 3). The appearance was consistent with chronic infarcts within the posterior circulation. Given the patient’s history of trauma and headache and the young age of onset, this was most likely secondary to vertebral artery dissection.

### 3.4. Famous Faces

She identified none of the 10 faces. However, her scanning of them was qualitatively as expected. Figure 5 shows a typical example of the patient’s scanpath, here to face of the cricketer Donald Bradman. They generally followed the familiar “inverted triangle” pattern. Figure 6 shows the number of fixations and fixation durations for the individual images. A repeated measures one-way ANOVA (using Šidák’s correction for multiple comparisons) found a highly significant difference between AOIs (F = 16.71, *p* < 0.001), with the counts for both the eyes and mouth being significantly different from the nose (*p* < 0.001). The pattern for fixation duration was similar but fell just short of significance (*p* = 0.055).

### 3.5. Emotional Faces

Unlike the complete failure seen in the effort to identify famous faces, emotion recognition performance varied with the emotion shown (Figure 7).

She was correct eight out of eight times for happy faces and five out of eight for surprise, which were statistically significant (binomial test, *p* < 0.05). Responses to other emotions were nonsignificant. Again, scanning was qualitatively normal, with fixations to the eyes and mouth being predominant. Analysis of fixation counts to the eye, nose, and mouth AOIs (Figure 8a) showed main effects for emotion (df = 6, F(6, 147) = 10.11, *p* < 0.0001, 7.81% of variance), facial features (df = 2, F(2, 147) = 251.6, *p* < 0.0001, 64.81% of variance), and the interaction between them (df = 12, F(12, 147) = 5.46, *p* < 0.0001, 8.44% of variance). For fixation duration (Figure 8b), the main effect of facial feature (df = 2, F(2, 147)—249.7, *p* < 0.0001, 72.44% of variance) and the interaction term (df = 12, F(12, 147) = 2.654, *p* = 0.003, 4.6% of variance) were significant, but emotion fell short (df = 6, F(6, 147) = 1.879, *p* = 0.088, 1.64% of variance). Analysis of the multiple comparisons with Tukey’s multiple comparison test for the ANOVAs found that the significant differences were for fixation counts were for the eyes: fear vs. happy, *p* < 0.05, and mouth: disgust vs. happy, *p* < 0.0001, disgust vs. sad, *p* < 0.01, fear vs. happy, *p* < 0.0001, happy vs. sad, *p* < 0.0001, happy vs. surprise, *p* < 0.0001, happy vs. neutral, *p* < 0.0001. For fixation duration, the significant multiple comparisons were: for the eyes, fear vs. happy, *p* < 0.05; for the mouth, disgust vs. happy, *p* < 0.01, disgust vs. sad, *p* < 0.01, disgust vs. surprise, *p* < 0.05 and disgust vs. neutral, *p* < 0.05).

### 3.6. Colour Plates

As was also observed in her neuropsychological assessment, she was unable to identify any of the plates, in contrast to her performance on the FM-15 colour test. As shown in Figure 9a, a control subject needs only a few fixations to correctly identify a plate. Even numerous fixations on salient portions of the plate were insufficient for the patient to identify it (Figure 9b). When she was allowed to trace the figure manually and given a minute to do so (Figure 9c), she could still not identify it, although her fixations accurately mapped out its form.

### 3.7. Navon Figures, Plain and “Dotted”

For the plain Navon figures, both local and global letters were identified correctly 20/20 times. For the dotted figures, the local letter was identified correctly 19/20 times, but the global letter was not identified correctly in any trial. For analysis, images were divided into salient and non-salient regions, the former being the small letters that make up the larger, global one, and the latter being the black dots or white space. The number of fixations to the salient global letter and the non-salient surround, either dotted or undotted, was tallied, and their cumulative duration to each region was calculated. An example is shown in Figure 10.

Figure 11a,b show the fixation counts (left) and fixation durations (right) for the conventional and dotted Navon types. Two-way repeated measures analyses of variance (rmANOVAs) were run for both fixation count and duration. The ANOVA for fixation count found a significant main effect for salience (the Navon character itself versus the background, whether dotted or plain) (df = 1, F(1, 19) = 264.1, *p* < 0.0001, 74.8% of variance). “Dottedness” fell somewhat short of significance (df = 1, F(1, 19) = 3.651, *p* = 0.071, 1.07% of variance), and the interaction was clearly nonsignificant (df = 1, F(1, 19) = 0.633, *p* = 0.436, 0.23% of variance). Similarly, the ANOVA for duration found a main effect for salience (df = 1, F(1,19) = 237.9, *p* < 0.0001, 77.3% of variance), no effect of dottedness (df = 1, F(1, 19) = 1.9534, *p* = 0.178, 0.36% of variance), and a trend towards an interaction between them (df = 1, F(1,19) = 3.615, *p* = 0.073, 1.39% of variance). Thus, gaze behaviour towards the Navon letters themselves was largely unaffected by the presence of a dotted surround, while there was a tendency for the cumulative time spent looking at the dotted surround to be longer than that spent looking at blank off-target areas. The modest effect of the dots on fixation behaviour contrasted with their causing a complete failure of global letter recognition.

## 4. Discussion

This paper presents, for the first time, the performance of someone with ventral simultanagnosia on a series of perceptual tasks requiring the global integration of local information, with concurrent recording of gaze. This allows us to look for relationships between the acquisition of visual information and its processing into a coherent percept. Our main hypothesis was not supported: even when task performance was greatly impaired, this was not reflected in gaze distribution. We have, however, considerably extended our understanding of the limits of holistic processing in this rare condition.

Given that feature integration is impaired but not absent in ventral simultanagnosia, we can thus look for commonalities across tasks. Ventral simultanagnosia leads to less severe deficits than the dorsal form [5,6,7]. Farah described the key difference: “In dorsal simultanagnosia, perception is piecemeal in that it is limited to a single object or visual gestalt, without awareness of the presence or absence of other stimuli. In ventral simultanagnosia, *recognition* is piecemeal, that is, limited to one object at a time, although, in contrast to dorsal simultanagnosia, other objects are *seen* [6].” That is, in dorsal simultanagnosia, only one element in the visual environment at a time elicits a response. In the ventral form, multiple elements may be responded to (e.g., as in a counting task) but without reaching conscious awareness. Analysing gaze can provide evidence of which parts of the scene are attended to, whether consciously or otherwise. Thus, during the patient’s scanning of the Cookie Theft picture (Figure 4), she fixated on areas relevant to the scenario, both those she mentioned and those she did not recognise. While her scanpath resembled that of a patient with dorsal simultanagnosia [24], unlike him, she was able to summarise key activities in the scene.

The two face-processing tasks had different relationships to gaze behaviour. At a qualitative level, fixations on famous faces followed the familiar inverted triangle [34,35], unlike the aberrant scanning seen in an acquired prosopagnosia patient [25]. The latter authors attributed their patient’s inability to recognise faces to his failure to fixate the inner, salient elements of faces, thus attributing perceptual impairment to abnormal information acquisition. In contrast, our patient at least qualitatively took in information from those salient features but completely failed to integrate them into an identifiable percept. Consistent with this, Van Belle et al. [36] found in a masking experiment that a prosopagnosia patient processed faces feature by feature, showing no evidence of the holistic integration needed for recognition.

In contrast, the identification of facial emotion depends less upon holistic processing. Wegrzyn et al. [37], also using the Ekman FACT database, found that various emotions were expressed differently by the inner face features. The recognition of sadness and fear depended most on the eyes, while disgust and happiness depended on the mouth. They also noted that anger and disgust, as well as fear and surprise, were frequently confused, attributing this to how the areas around the eyes express these emotions. Our patient identified happiness on all trials, saying that she could do so because she always looked at the mouth and so readily saw a smile. If we consider the frequently confused anger–disgust and fear–surprise pairs, then if the corresponding responses are combined, they would also have reached significance. This is consistent with the fixation patterns shown in Figure 8, where both the eyes and mouth attracted attention, and the pairwise multiple comparison tests, where happiness and disgust were frequently significantly different from the other emotions.

Analysis of the fixation patterns made using the pseudoisochromatic (“Ishihara”) plates is descriptive but nonetheless informative. Some patients with dorsal simultanagnosia were unable to identify the digits while successfully identifying the individual colours of the circles [16], but another similar case could identify them [17]. One case of ventral simultanagnosia evaluated on the task also failed at it and, unlike our patient, could not trace the figure with his finger [4]. Our patient performed normally on the FM-15 colour test but could not integrate her repeated fixations on the digit on the colour plate (e.g., Figure 9b) into a percept. When given sufficient time, she could trace the numerals with her finger, mapping out her finger’s movement along the digit (Figure 9c). This is consistent with Milner and Goodale’s framework of vision for perception and vision for action [38], whereby the ventral stream provides the necessary information for recognition while interacting with the dorsal stream to guide motor behaviour. This interaction may be occurring in a region in the posterior parietal cortex [39]. As this region was not damaged in our patient, local information was thus available to drive her motor system, guiding her finger. However, due to damage to the temporooccipital cortex (area TO), the limited holistic integration mechanisms available could not overcome the competition from the surrounding circles to allow her to perceive the digit that her finger traced.

The final task discussed is the Navon global/local letter identification task [12]. This task was designed to assess the ability to process both global and local information. Navon found that in normal individuals, the global form takes precedence, which should make it a particular challenge for individuals with simultanagnosia. As described earlier, an individual with dorsal simultanagnosia performed normally on local letter identification but was impaired when naming global characters, even though their gaze covered the entire character [16]. Showing the potential utility of proprioception in form identification, another similar patient could identify only the global letter with her eyes closed while her finger was moved passively over it [19]. This contrasted with our patient’s performance when tracing a figure on one of the colour plates. However, her performance on conventional Navon characters was almost perfect, which is consistent with the limited preservation of global processing in ventral simultanagnosia. Her failure to name the colour plates motivated us to modify the Navon characters by adding dotted surrounds analogous to those in the colour plates, providing competition for attention in a way similar to the effect of the coloured surrounds in the Ishihara plates. These novel stimuli had a profound effect on her performance. Her recognition of the local letters fell only from 20/20 to 19/20, but recognition of the global letters fell from 20/20 to 0/20. This might have been because the added distractors drew her gaze away from the global letter, but our eye movement recordings refuted this possibility. As Figure 11 shows, confirmed by subsequent ANOVAs, gaze was nearly unaffected by the addition of the surround (contradicting our hypothesis), but perceptually, global feature integration was abolished. Considering this and our patient’s inability to perceive any of the colour plate digits, it is clear that local elements defined by achromatic form and those defined by colour become impossible to integrate when faced with competing distractors in this individual with ventral simultanagnosia, even though fixations continue to be directed to the salient elements of the stimulus.

One caveat that must be raised is the role of our patient’s visual field deficit. Ideally, a control group with similar field loss, as might occur in optic nerve or retinal disease, would have addressed this issue. In its absence, we can point to the fact that on several tasks, her impairment was the same regardless of viewing distance, but whether field loss contributed to impaired performance cannot be ruled out. The other limitation is, of course, that this is a report on a single patient, so any generalisations about the findings can only be tentative. It is hoped that if further patients come to light, the hypotheses tested here could be further examined. Whether any of the deficits observed could be ameliorated through perceptual training might also be examined.

## 5. Conclusions

As the rarer variant of a rare condition, ventral simultanagnosia has been reported infrequently, generally in reports with a narrow focus. Using for the first time an array of perceptual tasks with concurrent gaze tracking, we have demonstrated that while gaze was qualitatively normal in both face recognition and emotion recognition, failure was total for the former task and partially successful for the latter. We also demonstrated for the first time that a task performed successfully—recognition of the global letters in Navon stimuli—could be made impossible by the addition of distractors but that these distractors had minimal effects on gaze. This likely explained the patient’s failure on the Ishihara plates, as the dots that formed the digits to be identified were similarly surrounded by distractors. This finding further suggests that in such patients, even capabilities that remain might be disrupted by changes in their visual environment, a possibility that could have practical implications. This general absence of abnormal gaze behaviour thus resolves the question raised by abnormal perception as to whether the deficit lies in an inability to acquire or to process the relevant information, showing that often, it lies in processing.

## Figures and Tables

**Figure 1 neurolint-17-00105-f001:**
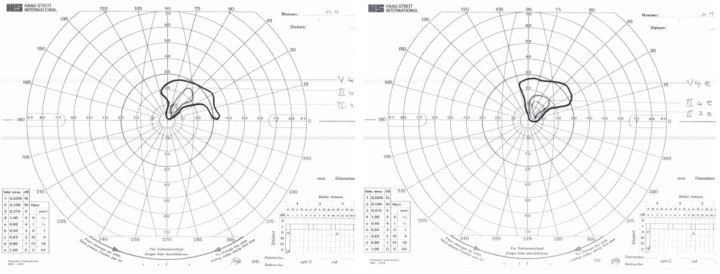
Visual fields of MH recorded with Goldmann perimetry (23 April 1976).

**Figure 2 neurolint-17-00105-f002:**
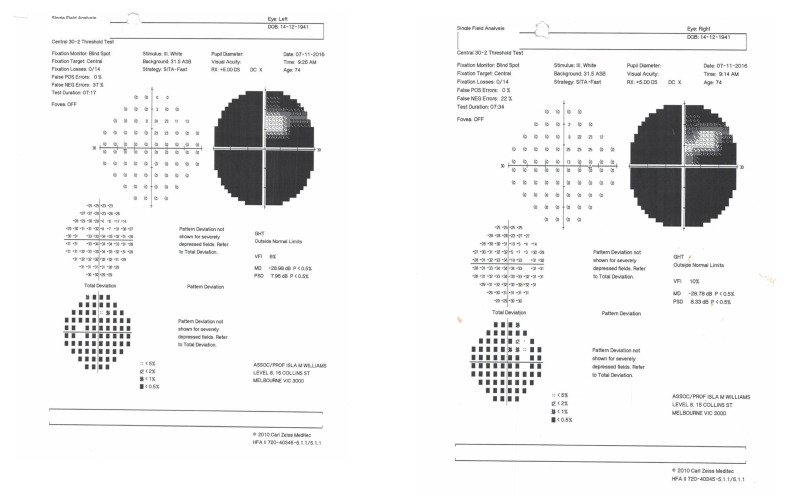
Visual fields of MH recorded with a Humphrey automated perimeter (7 November 2016).

**Figure 3 neurolint-17-00105-f003:**
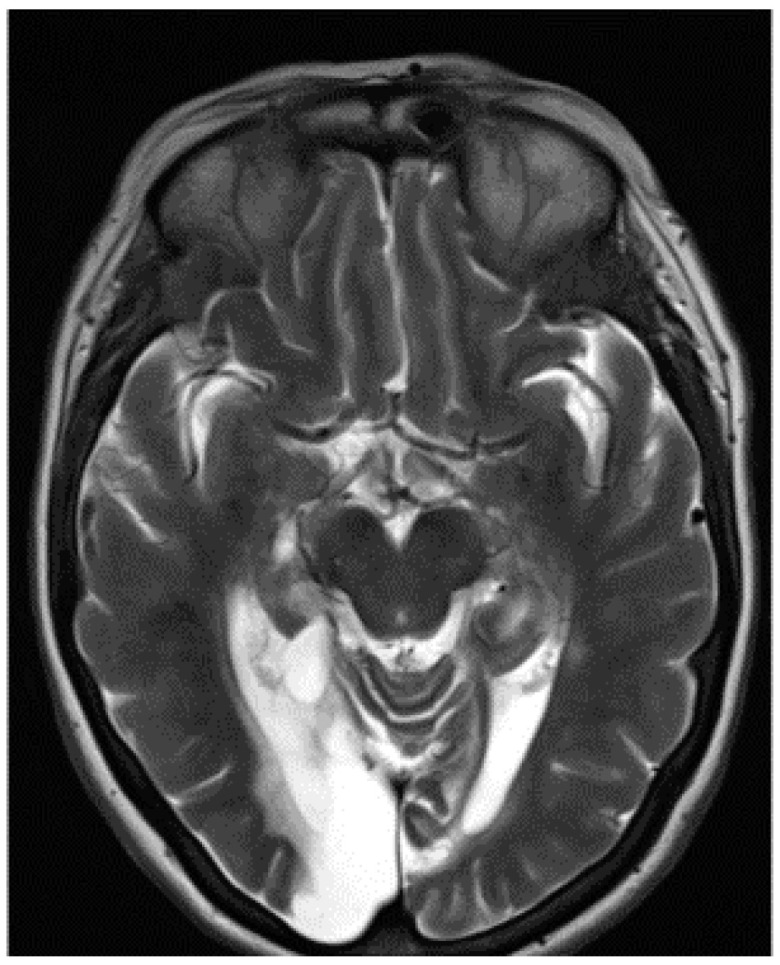
MRI scan of the brain showing encephalomalacia in the occipital and posterior temporal lobes bilaterally and the left superior cerebellum with bilateral thalamic infarcts.

**Figure 4 neurolint-17-00105-f004:**
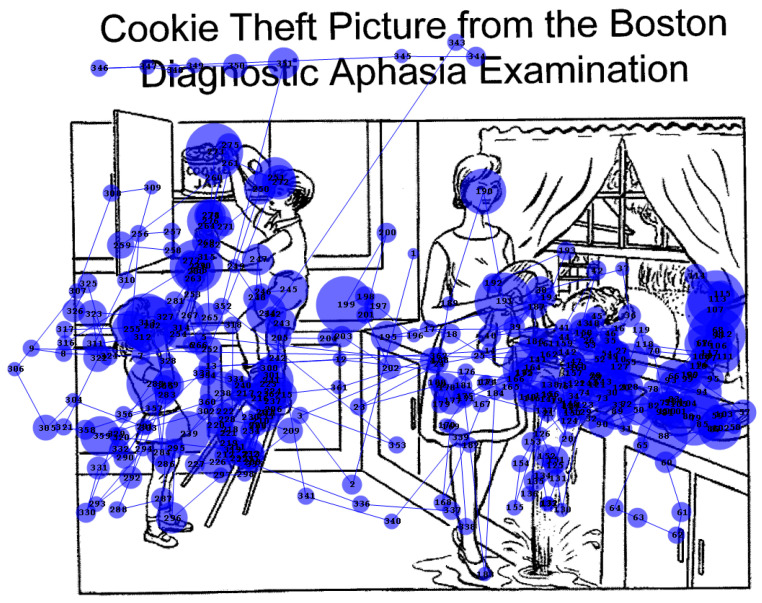
Scanpath for Cookie Theft picture, showing fixations of salient features. Used with permission.

**Figure 5 neurolint-17-00105-f005:**
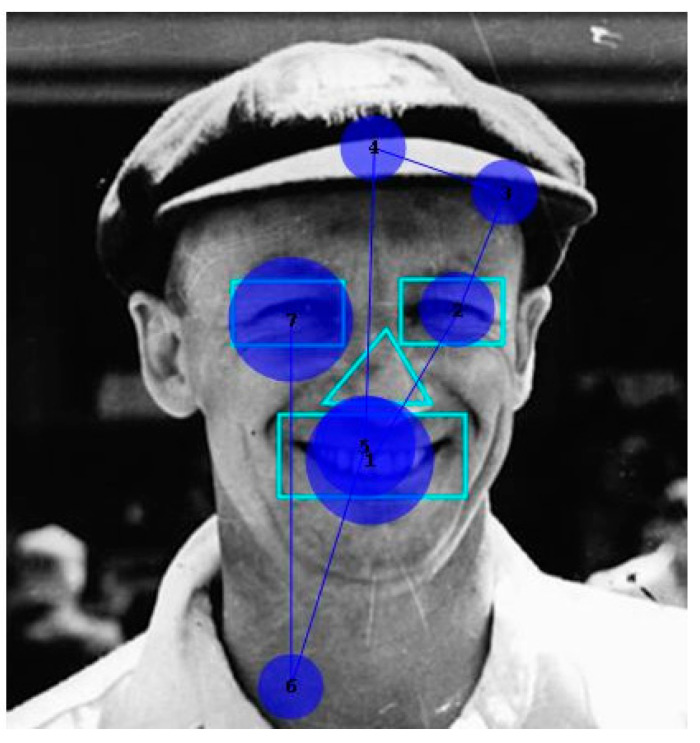
Typical inverted triangle fixation pattern produced when viewing cricketer Don Bradman. Circle size corresponds to fixation duration.

**Figure 6 neurolint-17-00105-f006:**
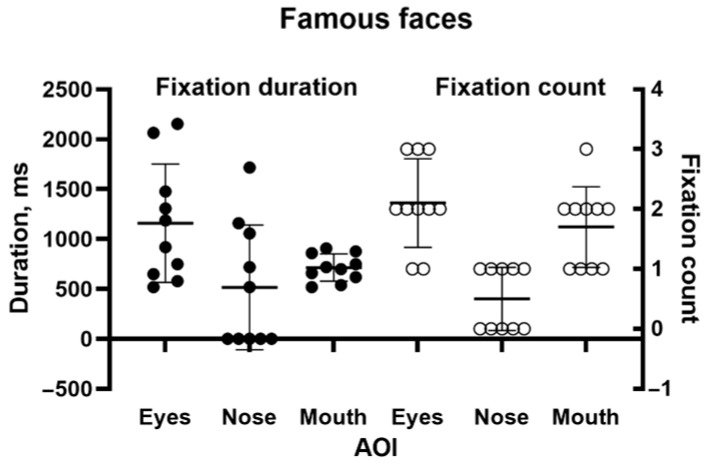
Duration (left) and number of fixations (right) for inner facial features during the famous face recognition task.

**Figure 7 neurolint-17-00105-f007:**
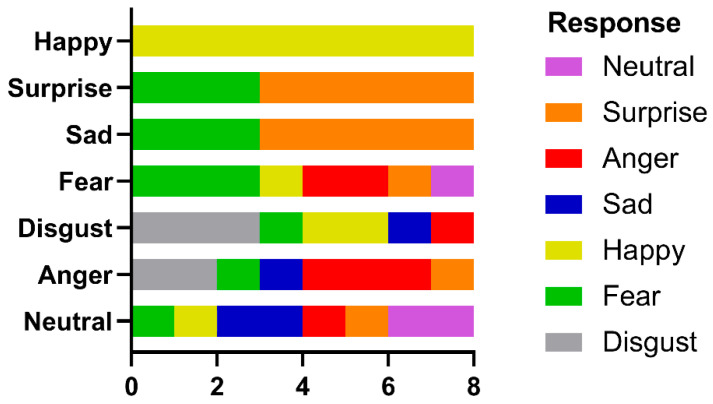
Identification performance in the recognition of facial emotion.

**Figure 8 neurolint-17-00105-f008:**
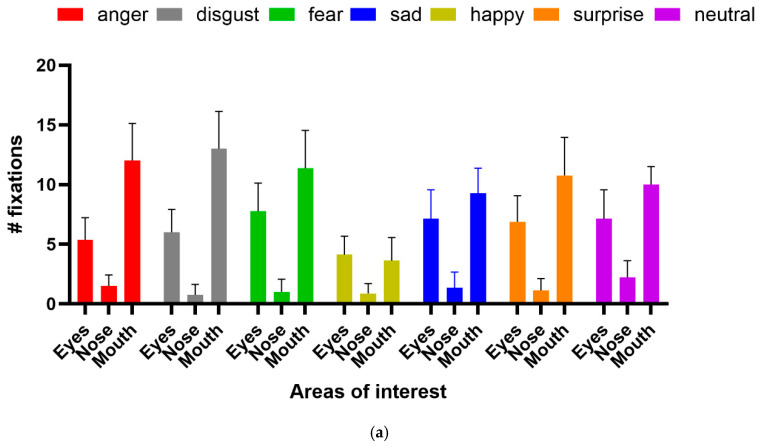
(**a**) Fixation count and (**b**) fixation duration for facial emotion recognition task.

**Figure 9 neurolint-17-00105-f009:**
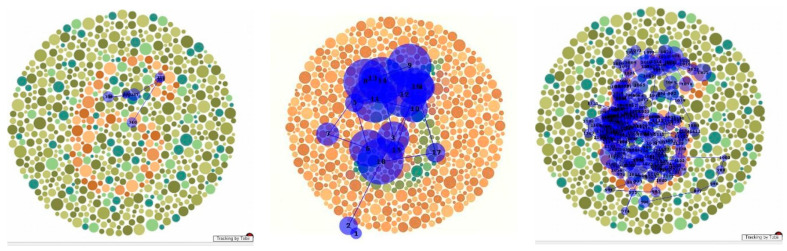
Scanpath during viewing of Ishihara plates by (**a**) normal subject, (**b**) MH, and (**c**) MH when tracing figure with her finger.

**Figure 10 neurolint-17-00105-f010:**
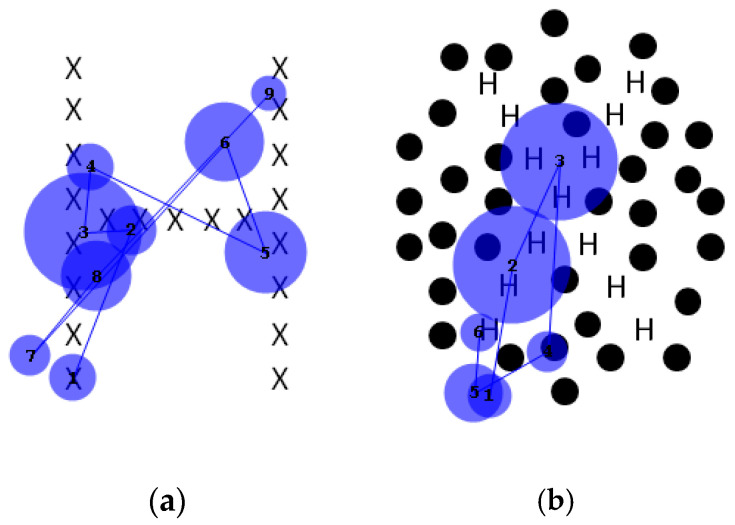
Sample scanpaths of (**a**) conventional and (**b**) “dotted” Navon figures, showing fixations nearly always to salient elements (i.e., the small letters comprising the global letters).

**Figure 11 neurolint-17-00105-f011:**
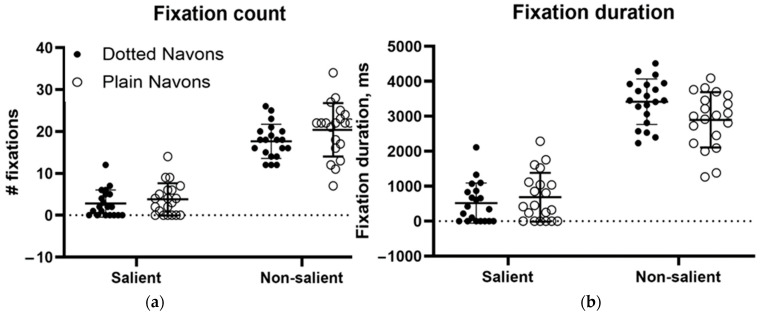
(**a**) Fixation count and (**b**) fixation duration to the salient (i.e., letters) and non-salient (i.e., background) elements of plain and “dotted Navon figures.

## Data Availability

Data that would not identify the subject of the study (i.e., not-clinical) can be made available on request.

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
