# Peer review of "Visual Perception and Fixation Patterns in an Individual with Ventral Simultanagnosia, Integrative Agnosia and Bilateral Visual Field Loss"

_2035-8377, 2025, doi:10.3390/neurolint17070105_

Round 1

Reviewer 1 Report

Comments and Suggestions for Authors

The manuscript presents a detailed case study of a patient with ventral simultanagnosia, integrative agnosia, and bilateral visual field loss, examining gaze behavior and perceptual deficits across multiple tasks.

While the study is well-structured and provides valuable insights into the dissociation between visual information acquisition and perception, there are some major weaknesses need addressing before publication.

Below are specific feedback and aconstructive recommendations.

  1. The abstract is poorly written. To clarify the abstract, add a sentence like, "We hypothesized that ventral simultanagnosia involves intact visual acquisition but impaired perceptual integration."
  2. The abstract lists findings but does not emphasize their novelty or implications.
  3. Highlight the key takeaway (e.g., "This study provides the first evidence that...").
  4. The abstract mentions performance differences (e.g., 19/20 vs. 0/20 in Navon tasks) but lacks statistical significance.
  5. Include p-values or effect sizes.
  6. The introduction could be condensed by removing redundant descriptions of prior case studies. Focus on key studies that directly inform the current work (e.g., Farah’s distinction between dorsal/ventral forms).
  7. The introduction doesn’t justify why famous faces, Ishihara plates, and Navon figures were chosen.
  8. Link task selection to the hypothesis (e.g., "To test holistic vs. feature-based processing, we selected tasks requiring integration of local/global features").
  9. The introduction doesn’t clearly mention how this study add to our existing knowledge
  10. Add a paragraph summarizing the unique contributions (e.g., first comprehensive gaze analysis in ventral simultanagnosia).
  11. The study uses two different eye-trackers (Tobii 1750 and Eyelink 1000) without justification.
  12. Explain why different systems were used (e.g., hardware availability) and address potential confounds (e.g., differences in sampling rates).
  13. The study relies on qualitative comparisons to "normal" gaze patterns but does not include a matched control group.
  14. Add a small control cohort or cite normative data for fixation patterns in healthy subjects.
  15. Specify statistical methods (e.g., Tukey’s HSD for post-hoc tests) and alpha correction (e.g., Bonferroni).
  16. The study concludes that gaze was "normal" in famous face tasks, but without controls, this is speculative.
  17. Mention findings as "qualitatively similar to expected patterns" rather than "normal."
  18. Some results (e.g., fixation duration in emotional faces) are described as "falling short of significance" without effect sizes.
  19. Figures 6a/b and 8a/b could be merged for conciseness. Combine fixation count/duration into single panels with subplots.
  20. The discussion extrapolates findings to "ventral simultanagnosia" broadly, but this is a single case study.
  21. Emphasize limitations (e.g., "These results may not generalize to all ventral simultanagnosia cases").
  22. The study dismisses visual field loss as a confound but does not empirically rule it out.
  23. Discuss how field loss might interact with simultanagnosia (e.g., scotomas disrupting integration).
  24. The discussion doesn’t propose a neural model for why gaze is intact but perception fails.
  25. Link findings to ventral stream theories (e.g., impaired V4/IT integration).
  26. The conclusion does not highlight clinical or theoretical implications. Add takeaways (e.g., "Future work should explore whether gaze training improves integration in simultanagnosia").
  27. Some grammar issues are present such as inconsistent tense. Sentences shift between past and present tense (e.g., "MH described difficulty" vs. "This study demonstrates").

Reviewer 2 Report

Comments and Suggestions for Authors

This paper examines the phenomenon of simultanagnosia, a neurological condition where patients have a visual agnosia (i.e. inability to form a normal percept despite normal primary sensation) such that the defect is in the simultaneous integration of a scene’s spatially disparate elements. The paper wants to focus on the subtype of ventral simultanagnosia, when brain dysfunction is implicated in the inferior and posterior regions of the brain’s cerebral hemispheres. The paper states its purpose is to address the specific question as to how much and in what ways ventral simultanagnosia might stem from how changes in eye movements affect how a patient scans a visual scene to take in information versus stemming from changes in the subsequent processing of visual information after that information has been acquired. The primary method used in this paper is to present a case study and describe characteristics of the patient’s neuropsychological profile and use eye tracking to show how the patient scans various presented stimuli.

The general purpose of the paper is worthy and interesting.  The paper presents good references in the introduction and discussion and explains these papers well.  However, overall, the paper is seriously flawed and as it stands should not be published.

A case study of a single patient is a time-honored kind of publication in clinical neurology and especially in behavioral neurology. Such a category of publication is valuable since averaging across patients often removes the special and unique glimpse into cerebral operations that a good in-depth single patient can provide. However, in this paper, the paper drifts quite far from its stated purpose and loses much of the specialness that a case study can provide. The paper purports to focus on how eye movements have not been well studied but could still explain ventral simultanagnosia, which is an intriguing idea that deserves attention. Yet, about half of the introduction (lines 61-94, 111-151) seems irrelevant to this stated goal, going into studies and details that are not clearly related to the stated purpose of the paper. The discussion has similar problems.  Also, while some of the eye movement findings are interesting, the results and discussion reads more like a mere exposition of findings and lacks a coherent argument as to what these findings actually mean for the purported purpose of the paper.  The paper thus contains little argument as to what the reader now knows about simultanagnosia in general, or about eye movements in a case of ventral simultanagnosia, other than to state (results) and then restate (discussion) that this person yielded these data.  Indeed, much of the discussion again strays away from the purported main reason for the paper – how eye movements may explain ventral simultanagnosia – and instead simply states various non-eye-movement neuropsychological and other findings that occurred in this particular patient, again with little attempt to argue to the reader what any of this actually means.

In short, this paper fails in that it does not sufficiently allow itself to “be about” because it does not argue for ideas or conclusions, and it's focus is vague and meandering in that it neither introduces well nor delivers on the specific question(s) it claims to be addressing.

Comments on the Quality of English Language

The paper is generally well written, though the writing is marred by frequent typographical errors especially with punctuation and spacing. 

Round 2

Reviewer 1 Report

Comments and Suggestions for Authors

The authors have addressed my comments well.

Author Response

Thank you (and thanks for the very helpful initial review).

Reviewer 2 Report

Comments and Suggestions for Authors

This paper is longer and more expansive than needed in the topics that it ambitiously tries to cover, and in in the treatment of each topic, also in the details of the case reported, all in ways not relevant to the essential thrust of the paper.  The essential point of the paper is well articulated in the abstract (lines 39–44).  The points made there are good, useful, and worthy of being in the literature.  Those points are what the reader should remember, and what the reader should feel is the reason for having read the paper.  Instead, a reader of this paper is going to find it hard to know why the paper was written or what they should think they know after having read it.  The introduction meanders onto topics of simultanagnosia that are interesting to those of us who like behavioral neurology and neuropsychology but really are not relevant to understanding the points articulated in lines 39–44.  Similarly, the extensive details of the case, even if factual and true and again interesting to those of us who like behavioral neurology and neuropsychology, nonetheless detract from a reader seeing and understanding the main points of the paper as articulated in lines 39–44, points that at best are difficult to see and at worst not possible to see throughout the results and discussion sections, and are barely articulated in the conclusion.  So, while I judge the paper as publishable, and while I can see that the authors appear to be collectively knowledgeable and articulate about behavioral neurology and neuropsychology topics, I am sad because this paper deserves to be a punchy, clear, to-the-point, memorable paper that readers will understand and want to cite, and instead it is unfocused and its main points are unlikely to be understood or remembered. 

Despite several revisions, the paper still contains a number of errors in punctuation, spacing, and it still contains typos.  It reads as if it has been loosely proofread, or has used different authors to write different sections.  (1) Some sentences are still missing a period at the end of the sentence (e.g. lines 73, 99).  (2) There is inconsistent spacing in that sometimes more than one space occurs after a bracket (e.g. line 193).  (3) There is inconsistent spacing in that after an end-of-the-sentence period, there is sometimes 1 space, sometimes 2, sometimes more than 2; whatever the decision is going to be, it should be used consistently throughout the paper.  (4) There is inconsistent spacing before and after a numeral such as when describing dimensions (e.g. [numeral]x[numeral] versus [numeral] x [numeral]) or when a numeral is followed by an quantity abbreviation (e.g. [numeral][abbrev] versus [numeral] [abbrev]).  The paper should pick one style and stick with it.  (5) There is inconsistent use of an extra line between paragraphs.  (6) There is inconsistent use of paragraphs that began with indentation and followed by no line break (I saw this in the entire paper before discussion) (though line breaks are sometimes inserted and sometimes not; should pick one style) and paragraphs that are always unindented (I saw this in the discussion). Unless such variation is asked for by the journal's standards, the paper should pick one style, and stick with it.  (7) There are multiple typos that should be corrected (e.g., lines 140, 381).  
